# Exploring Barriers to One Health Antimicrobial Stewardship in Sri Lanka: A Qualitative Study among Healthcare Professionals

**DOI:** 10.3390/antibiotics11070968

**Published:** 2022-07-19

**Authors:** Yasodhara Deepachandi Gunasekara, Tierney Kinnison, Sanda Arunika Kottawatta, Ruwani Sagarika Kalupahana, Ayona Silva-Fletcher

**Affiliations:** 1Department of Veterinary Public Health and Pharmacology, Faculty of Veterinary Medicine and Animal Science, University of Peradeniya, Peradeniya 20400, Sri Lanka; sandavphp@gmail.com (S.A.K.); ruwanikalupahana@yahoo.com (R.S.K.); 2Royal Veterinary College, University of London, London NW1 0TU, UK; tkinnison@rvc.ac.uk (T.K.); asilvafletcher@rvc.ac.uk (A.S.-F.)

**Keywords:** antimicrobial resistance, antimicrobial stewardship, health care professional, one health approach, Sri Lanka

## Abstract

Antimicrobial resistance (AMR) is a global health threat, but little is known about the perceptions regarding antimicrobials and AMR among healthcare professionals in Sri Lanka. This research aimed to take a One Health approach to explore the knowledge, attitudes and perceptions of antibiotic stewardship and AMR among healthcare professionals in Sri Lanka. A qualitative study, using telephone interviews, allowing for an in-depth exploration of attitudes, beliefs and perspectives was conducted. Healthcare professionals from both the medical and veterinary sectors were included (*n* = 29). Interviews were conducted by an independent interviewer and were audio-recorded and transcribed. Conventional qualitative content analysis was undertaken. Four main categories were identified: (1) understanding of AMR and observing AMR, (2) barriers to antimicrobial stewardship, (3) personal factors in, and as a result of, inappropriate antibiotic usage and (4) how to tackle AMR. Healthcare professionals showed poor awareness regarding the spread of AMR and identified inappropriate prescribing behaviours by their inter- and intra-professional colleagues. Patient demands and the influence of pharmaceutical companies were factors contributing to poor prescribing behaviour. Suggestions for the future are stricter regulation of AMR control policy, effective government involvement, and awareness campaigns for healthcare professionals and the public.

## 1. Introduction

Antimicrobial resistance (AMR) is becoming a major public and animal health issue worldwide. It raises human and animal morbidity and mortality rates while increasing the prevalence of AMR organisms in the environment [1]. Although several factors contribute to the rise of AMR, overuse and misuse of antimicrobials is identified as one of the major causes of AMR [2]. For the purpose of this study, personnel who had experience and knowledge regarding antibiotics and AMR, namely, physicians, veterinarians, dentists, nurses, pharmacists, medical laboratory technologists, public health inspectors, midwives and livestock development officers, were categorized under healthcare professionals. In Sri Lanka, only physicians, dentists and veterinarians are known by this term; however, in other countries, healthcare professionals could include other groups.

According to the World Health Organisation (WHO), antimicrobial stewardship is defined as “A coherent set of actions which promote the responsible use of antimicrobials. This definition can be applied to actions at the individual level as well as national and global level, and across human health, animal health and the environment” [3]. In 2015, the WHO announced a global action plan to tackle AMR which included antimicrobial stewardship [4]. With the support of the WHO country office, Sri Lanka announced a National Strategic Plan to mitigate the AMR issue using an antibiotic stewardship programme with a 2-year milestone [5]. In addition, the Sri Lankan College of Microbiologists started two projects which were aimed at national surveillance of antimicrobial resistance in Sri Lanka [6]. The Antibiotic Resistance Surveillance Project was started in 2009, and under this project, blood and culture isolates are investigated, while urine culture isolates are investigated under the National Laboratory Based Surveillance of Antimicrobial Resistance of significant urine culture isolates [7,8]. In 2016, the Sri Lankan college of Microbiologists, in collaboration with other professional collages of healthcare and The Ministry of Health, Nutrition and Indigenous Medicine of Sri Lanka, announced national guidelines for the empirical and prophylactic use of antimicrobials [9].

Gaps still exist in the knowledge, attitudes and perceptions (KAP) of AMR among healthcare professionals, and it has been suggested that it is necessary to improve the quality of teaching materials in the training of health professionals [10]. The antimicrobial prescribing behaviour of healthcare professionals—namely, physicians, dentists and veterinarians—may be directly related to the antimicrobial consumption pattern of citizens. Professionals who work in healthcare services have a responsibility to convey their knowledge regarding antibiotics and AMR to their patients and the general public. In addition, the failure to emphasise the importance of AMR to policymakers and responsible authorities by healthcare professionals contributes to the establishment of poor new laws and unsuccessful implementation of existing laws to tackle AMR [11].

Published literature supports the notion of antibiotic overprescribing practises. For example, the majority of doctors surveyed in one hospital in Lima, Peru (72%) ‘agreed’ that they overused antibiotics [12]. Irrational antibiotic prescribing patterns were also reported in a study which was conducted among doctors working in a public hospital in Sierra Leone [13]. In Pakistan, one study suggested that patients’ expectations of antibiotics put pressure on doctors and were one reason for their prescribing behaviours [14]. Research has also highlighted a lack of knowledge of AMR, including a study in DR Congo where, although the majority of medical practitioners sampled (77.2%) said that they were confident about their knowledge of antibiotics, there was no statistically significant correlation between their mean knowledge score and confidence level [15].

Other studies have demonstrated that practitioners do not feel AMR is a problem, or highlight the lack of impact practitioners believe they have on the situation. A group of primary care physicians who participated in a qualitative study in Spain felt that AMR was not a problem at a community level [7]. General dental practitioners working in Wales, UK thought that their contribution to the development of AMR was far less or negligible compared to that of general medical practitioners [8].

In Sri Lanka, a few studies have been conducted to evaluate KAP on AMR among healthcare professionals [16,17,18]. The results are similar to the previous literature cited, demonstrating a lack of understanding of fundamental aspects of AMR. One study reported that 40% of trainee nurses who participated in the study thought that taking antibiotics would help to prevent colds [16], and AMR knowledge of Sri Lankan pharmacy students was identified as being less than Australian pharmacy students [19]. A further study reported that nearly half of medical practitioners (46.6%) prescribed antibiotics for fungal and parasitic infections [20].

While KAP studies can provide a quantitative overview of knowledge, the potential differences in understanding within different groups of professionals and the complexity of the issue warrants in-depth investigation of KAP on AMR among healthcare professionals. Although our understanding of KAP on AMR among medical healthcare professionals is beginning to increase, KAP on AMR among veterinary healthcare professionals in Sri Lanka has yet to be researched. Bacteria are not limited to one species and therefore the human and animal sector use many of the same antibiotics, both contributing to the accumulation of residues in the environment. Hence, understanding KAP on AMR transmission within the ecosystem among all health service personnel, including both the human and veterinary fields, is important. In order to better establish AMR mitigating programmes, multi-sector collaboration is needed. It is not only important to evaluate KAP among antibiotic prescribers (Physicians and Veterinarians), but also within professions, occupations and groups who work alongside them. Subsequently, allied healthcare professionals should also be evaluated within the research. In Sri Lanka, this would relate to nurses and pharmacists from the human healthcare field, and from the veterinary field, livestock development instructors who advise farmers on farm animal management, disease control and prevention methods. All of these groups potentially have a role to play in a One Health approach to antimicrobial stewardship and are therefore important stakeholders in research on this topic.

Studies that treat a country’s population as homogenous may also lose the depth of understanding of AMR KAPs. Within Sri Lanka, animal and human healthcare facilities are unequally distributed throughout the country, and rural areas have fewer healthcare facilities [21]. Furthermore, higher-grade hospitals, diagnostic facilities and key healthcare professionals are concentrated in urban cities.

The focus of this research was to take a One Health approach to explore the social dimensions of antibiotic stewardship and KAP regarding AMR among Sri Lankan healthcare professionals from rural and urban areas. This required a qualitative study methodology, which allows for an in-depth assessment of knowledge, attitudes, beliefs and perspectives. It was anticipated that exploring this question through the One Health lens by incorporating both the human and animal sectors will help to guide the development of interventions to decrease AMR. This study was conducted as part of a larger study investigating the potential role of wildlife in AMR and ecosystem contamination in Sri Lanka.

## 2. Results

Overall, 29 healthcare professionals participated in the study; 20 were from an urban area and 9 from a rural area, with a total of 19 human healthcare professionals and 10 veterinary healthcare professionals. The demographic characteristics of the participants can be seen in Table 1. Participants were given identity codes to aid anonymity and to demonstrate the speaker of quotes throughout the results. In the following sections, the categories and sub-categories developed will be explored. These sections are summarised in Table 2.

### 2.1. Understanding and Observing AMR

#### 2.1.1. AMR Knowledge and Experience Acquisition

Almost all the interviewed health service personnel said that they were aware of AMR. The majority acquired their knowledge regarding AMR when they obtained their tertiary education such as an undergraduate degree or diploma, while also reporting that their professional experience helped them to develop their knowledge. Some participants also discussed obtaining knowledge via media and continuing self-study.

*“As health professionals, we were taught microbiology. But with time, the theories that were taught eventually have faded away from our minds. However, from the news and other discussions in society, I get some information regarding antimicrobial resistance”*.[UML1]

In human healthcare, participants observed neonatal deaths and urinary tract infections as consequences of AMR. In animals, the veterinary healthcare professionals described experiences of mastitis, swine fever, urinary tract infections and poultry disease. Participants also acknowledged their understanding of AMR was changing but identified the struggle to adapt to new knowledge.

*“There is a myth called ‘prevention of secondary infection’, which was taught in our university. We were misled, therefore, still, I use antibiotics to prevent secondary infection even in viral diseases based on that theory. In that case, I opt to use the strongest antibiotic I got. I think this is a big mistake made by practitioners”*.[RVE]

Inaccuracies in understanding were identified from the transcripts. For example, some recognised non-bacterial aetiological diseases also as a consequence of AMR. Inaccuracies in understanding and behaviour were also highlighted by the participants regarding their colleagues; one participant [RHD1] criticised physicians who prescribed antibiotics for viral infections.

*“Currently with the involvement of private pharmaceutical companies and private vets farmers use Gumboro plus [Vaccine] against Gumboro disease [Viral diseases]. I think in the future there will not be any [antibiotic] treatment for Gumboro and many birds will die as a consequence”*.[ULD1]

Participants provided additional examples of other professionals prescribing inappropriately and contributing to the AMR issue. The majority of healthcare professionals mentioned that other physicians who were operating private practises/private medical centres prescribe higher-order antibiotics and were trying to prescribe medicines including antibiotics that could cure diseases within a shorter duration and/or after taking a few doses. As a result, several participants used the term “one-shot doctor” [RVE1 &RHD1] to characterise these sorts of practitioners.

One livestock development officer placed the blame for poor prescribing behaviour on veterinarians.

*“Especially because of ignorance. Avoiding necessary steps is a problem. Some vets do not treat animals following all necessary steps; take steps towards higher antibiotics and proper care. Because of such Vets, animals reared by peoples as their livelihood, who can live for a long time happen to close their eyes pretty soon”*.[ULD2]

#### 2.1.2. Thoughts on AMR Development and Transmission

Healthcare professionals demonstrated a variable understanding of AMR development in the discussions. A very limited number of participants said that AMR develops in bacteria. However, most thought that AMR development takes place in the human body. Further, they thought the occurrence of AMR was related to the body immunity of the respective person.

*“When high antibiotics are given to patients as a practise, their bodies find it difficult to restore to good health because the bodies get used to it”*.[RNU2]

*“What causes is a person’s immunity and factors related, I think”*.[UNU4]

The majority of participants believed that humans who recurrently used antibiotics and/or who take the same antibiotic long-term will have the potential to develop AMR, with those who have taken antibiotics since childhood being more prone to developing AMR. On the other hand, some participants said AMR can be developed by the gene as an inherent characteristic of the human being.

*“Maybe because people grow with high antibiotics since their childhoods. With the way of the society, we live in, I feel that maybe the kids who are being born today might be inherited”*.[UNU1]

Another stated that the brand of antibiotic may have a relationship with the occurrence of AMR.

*“Sometimes people name a brand of drugs as the ‘best brand’. I have identified that it has its problems regarding the production of drugs. Just take Ciprofloxacin as an example, it is an Antibiotic and normally the best brand of this drug is XXX. XXX costs about Rs.80. Just assume that another brand of the same drug costs Rs.50. They are the same drug at different prices, yet the way each answer the body is so different. Then resistance is different”*.[UPM1]

Participants also mentioned that overuse and/or misuse of antibiotics, failure to take antibiotics on time and/or at recommended intervals trigger AMR.

The majority of participants were unable to explain the transmission routes of AMR. Many of them believed that AMR cannot be transmitted between humans and/or animals, since AMR developed in the human and/or animal body.

*“Do you think that there could be any possible way of getting it to humans and animals by getting collected in the environment? Nobody has ever told me such a thing. I don’t think so”*.[UHD1]

Even though some participants were able to identify animal-originated food and urine and/or faeces as a transmission route for AMR, almost all of them were not aware of the environmental route of transmission of AMR.

*“I’ve never heard that Antibiotic resistance could occur from water or soil. It is a rare occurrence; mainly Antibiotic resistance happens through food. Maybe these other ways could be counted under the same category too, I am not much aware”*.[UVE1]

### 2.2. Barriers to Antibiotic Stewardship

Human healthcare professionals and veterinary healthcare professionals reported similar barriers to antibiotic stewardship as explored below. Participants explored their challenges as well as the poor behaviour of ‘other’ members of their own and different groups.

#### 2.2.1. Patient and Farmer Demands for Antibiotics

Patients demanding antibiotics were identified as common by participants. Patients used different techniques to gain antibiotics, including re-using old prescriptions and providing the wrong clinical history.

*“There are some folks who lie to their doctors uttering fake symptoms relying on the thought that intake of more medicine can bring a healthy life. The doctor prescribes medicine for the symptoms a patient tells. This scenario is unlike in a hospital. In a hospital, a doctor gets the chance to visit the patients 2–3 times a day but in private dispensaries, a doctor meets his patients just once. A patient faking ailment could be easily understood in a hospital but it is unable to do so in a private dispensary or clinic”*.[RHD1]

Patients were thought to consider antibiotics as a quick fix and a natural expectation for various ailments.

*“A [teacher] once taught us to address this as ‘Capsule Syndrome’. You know, patients are not satisfied unless we prescribe them some kind of capsules”*.[UDD1]

Even those patients working within the healthcare system were reported as showing demanding behaviours by the prescribers.

In the livestock field, veterinarians had pressure from farmers to prescribe higher antibiotics.

*“Once when Swine fever was spreading among our pigs, some vets prescribed Augmentin for them. This happened in some other range, not in ours. Knowing this, a few guys from our range arrived in my place and we bickered on the very matter. They asked me to prescribe Augmentin and I kept on saying no. Augmentin is the main drug given to humans. That’s the only drug available even when a child gets pneumonia. There is the injection but it is inappropriate to give it to animals”*.[UVE3]

#### 2.2.2. Patient and Farmer Ability to Self-Treat and Disregard Healthcare Advice

Over-the-counter prescription of antibiotics was also identified as common practice by participants. Farmers treated livestock with over-the-counter medications based on their own diagnoses and/or imitated the treatments prescribed by veterinary doctors previously.

*“If the doctor leaves a used bottle of vaccines/drugs on the premises, some people are used to keeping the empty container with them and use that particular medicine as they wish. Mostly this happens in piggeries. People buy whatever medicines they expect would cure their animals and inject themselves”*.[ULD2]

During the interviews, it was highlighted that some farmers hesitated to follow professional recommendations. However, underlying reasons for this behaviour were not hypothesised.

*“We always advise farmers not to milk the cow or sell it for meat for a certain period of days if an injection was given. But some farmers do not seem to care about these health pieces of advice and they keep on doing whatever they want”*.[UVE2]

#### 2.2.3. Need to Maintain Professional Status and Client Base

The requirement to meet expectations in order to maintain professional standing and not lose customers (patients) was discussed as a reason for prescribing higher antibiotics at private clinics.

*“Whether one works in a government or private institution, he/she tries to continue and be successful at P.P (Private Practise). So for that, the doctor needs to have a good name among crowds. He cannot obtain this by prescribing normal antibiotics and tends to cure ailments with high antibiotics. Patients like this kind of doctor as what they expect are to get cured as soon as possible”*.[ULD1]

It was reported that patients were able to change doctors with ease should they not receive the treatment they expected, thus pressuring doctors to treat, or else simply move the issue onto another doctor.

*“Being a doctor if I don’t prescribe antibiotics people often say that It’s not worth getting treatment from me. They turn to the doctor who is prescribing it for them”*.[UHD4]

#### 2.2.4. Antibiotics as a Commodity

The majority of participants stated that multinational and national pharmaceutical companies had a significant influence on both the human and veterinary sectors.

*“Either pharmaceutical companies directly contact the farmers or farmers directly contact the pharmaceutical companies to decide to choice of antibiotic”*.[UVE4]

*“Medical Reps of companies promote it (antibiotics). These things really do happen”*.[UPM1]

The explanation for this was considered to be that they had to encourage antibiotics due to commercial commitments.

*“There are some agents in a few companies. Their high officials are vet surgeons who are passed out graduates of the Faculty of Veterinary, just like us. If they love the country’s future, they have a responsibility to take steps toward misuse of Antibiotics. But sadly, their business is marketing and having sales goals. So they release the drugs into the market in a desperate way”*.[UVE3]

The majority of participants stated that policymakers were unable to implement current laws and regulations. Pharmaceutical companies were also hypothesized to potentially provide them with a commission, which may be the reason behind the poor strength of rules and regulations.

*“The main problem occurs from private companies. Those companies obtain huge profits by selling those drugs. That becomes the main obstacle. Plus, there certainly could be parties like political ministers who benefit from commissions through these things, who knows? Sri Lankan medical workers are generally not genuine. Most of the time, our higher officials and some of the medical workers are money minded and very dirtily avaricious”*.[UVE3]

Poorly implemented rules were suggested to result in the introduction of low-quality pharmaceuticals into the market and hospitals. The misuse of antibiotics was also questioned and highlighted the antibiotic growth promoter which was legally banned.

*“There might be rules imposed over these matters, but as a person who is engaged in this field for years now, what I have understood is that the regulations are weak indeed. Some farmers use antibiotics just as how fertilisers and pesticides are used in farming. There are regulations but in a slumbering state. I think antibiotic growth promoters are being used recklessly despite there are regulations on antibiotic growth promoters”*.[UVE1]

#### 2.2.5. The Issue of Illegal Prescription

It was identified that professionally unqualified personnel prescribed antibiotics to patients in hospitals as well as in pharmacies where the medications are distributed. This situation was condemned by the participants.

*“There is a hospital attendant who is acting as Ayurvedic doctor and often prescribes antibiotics to patients”*.[UPM2]

*“Generally speaking, in Sri Lanka, 75% of the pharmacies are run by non- pharmacists. They don’t have any formal education background and even they don’t have a conscience”*.[UNU3]

### 2.3. Personal Factors in, and as a Result of, Inappropriate Antibiotic Usage

#### 2.3.1. Understanding the Patient and Farmer’s Perspectives

Reasons for patients’ and farmers’ behaviours regarding antibiotics which were deemed to be inappropriate were thought to include patient financial difficulties, as identified by a pharmacist who noted that patients altered the recommended dose as they could not afford the full dose.

Overprescribing was also performed to assist patients, for example, if physicians assumed their patients would have difficulty attaining the required antibiotics, due to the inability to return to the hospital for a second antibiotic or due to future complications.

*“I live in [area] which is a rural area. Imagine a mother coming from 10–12 km away from the town to treat her baby. They have to wait in the queue for some time. The doctor also might have a feeling about future complications that should be treated early to prevent further inconvenience faced by that poor mother”*.[RPM1]

Veterinarians also demonstrated understanding regarding farmers using antibiotics to treat their animals themselves, as they acknowledged difficulties in access to veterinarians throughout Sri Lanka.

*“If we consider the veterinary field, there are a limited number of government veterinary surgeon’s offices. It’s virtually impossible to handle all the issues that arise in this area by government vet office or veterinary investigation centre alone. The farmers were left with no choice other than to give some medicine available to save their flocks without seeking veterinary advice. Therefore, they try with all the medicines available”*.[UVE4]

Poor patient–prescriber relationships were identified during the interviews, limiting patients’ ability to understand antibiotics and AMR. It was mentioned that physicians were unable to explain the situation and aetiology of the disease to their patients. A reason behind this was a high patient volume in hospitals.

*“The doctors here in Sri Lanka are not used to explaining even in simple terms about the illness to their respective patients…. Some doctors clearly explain everything and how to intake the prescribed medicines. Yet some doctors do not do this… when a lack of communication takes place, most patients just change their doctor and carry on. If the same happens there too, he carries on with the same process so that a single patient changes 2–3 doctors for one ailment”*.[RDH1]

#### 2.3.2. Consequences of the Personal Role in the AMR Battle

Participants recognised poor behaviours as contributing to AMR as well as a subsequent personal inability to treat animals.

*“I got a very heartbreaking storyrecently happened to me. … resistance had been developed against all the antibiotics available. We didn’t have any antibiotic to treat that animal”*.[UVE4]

In an effort to maintain their ground on not prescribing or using antibiotics, participants reported disagreements with patients, including negative retorts such as being shouted at, as well as disagreements with other professionals.

*“Even though we quarrelled, I did not take steps to do the above, yet one of our vets did not hesitate to. From a drug company or some pharmacy, they have had obtained the necessary amount of Augmentin”*.[UVE3]

### 2.4. How to Tackle AMR

#### 2.4.1. Thoughts on a Personal Role in Tackling AMR

Almost all of the attendees agreed that combating AMR in Sri Lanka was a concerning issue. However, a dentistry practitioner stated that it was not a major concern in their field, and a livestock development officer believed that they were not responsible for combating AMR because they were not prescribing antibiotics.

*“This issue does not affect us. we do not prescribe Antibiotics for animals. What we do is guide people on how to give animals antibiotics properly and effectively only as per prescribed”*.[ULD2]

*“As dental surgeons, we use Antibiotics. But we do not use many varieties. So that I don’t think that this phenomenon might be happening large scale in our field”*.[UDD1]

#### 2.4.2. Potential Interventions to Tackle AMR

##### Empowering and Establishing the Rules and Regulations

The majority of participants suggested that legislation be strengthened to prevent over-the-counter prescriptions. Veterinary professionals recommended establishing a procedure to monitor animal-derived commodities.

*“There are rules and regulations. We always discuss these new ideas and proposals for making the system clear and strong. The problem is that all those confines to ‘discussions’, just ‘discussions. These discussions happen for a long time, but there is no change takes place”*.[UVE1]


*“Milk is not checked properly now in Sri Lanka. Maybe there are rules and regulations on these matters but they are not functioning correctly.”*
[ULD2]

##### Increasing Diagnostic Facilities and Human Resources

Both veterinary and human healthcare participants noted diagnostic uncertainty as one of the most prevalent issues they faced, suggesting the need for improvements and expanding diagnostic capabilities and human resources, especially in rural settings.

*“My clinic is located in the [area]. If I need to get a culture report I have to submit samples to [place]. But with my busy work schedule, it is almost impossible to do it in time. Therefore, it would cause more damage to the animal when the culture report arrives. So, it must minimise those types of difficulties”*.[RVE2]

##### Improving Farm Management Practises

Some veterinarians emphasised the value of biosecurity management on livestock farms as a possible approach to combat AMR.

*“it is important to implement biosecurity of the farm to prevent disease occurrence and spread”*.[UVE3]

##### Conducting Awareness Programmes

Healthcare professionals had variable thoughts on conducting awareness programs. The majority of the participants suggested that antimicrobial stewardship awareness programs should be conducted to sensitise the general public and healthcare professionals on antibiotics.

*“The very first thing must be spreading awareness among people. People must be convinced how important Antibiotics are”*.[UHD3]

Other participants felt that educating health care personnel should take priority over the general public, with one participant considering educating the public to be potentially dangerous.

*“Some people still use this knowledge for illegal causes. Just a little knowledge on any drug can make people go buy it as they wish”*.[UMF1]

Many participants believed that educating field officers and allied healthcare professionals was more necessary than educating doctors because doctors were already aware of the problem, while allied professionals would appreciate an increase in competence and confidence regarding the issue.

*“Many officials in this field provide us with limited knowledge about medicine. So how can we raise our voices when we see people trying to get drugs in various unaccepted ways? We might get the power to raise our voice if only we had a proper understanding of drugs, their uses and side effects either during our training period or at MOH levels. Bearing a little knowledge, we cannot do this, so I hope we get the chance to be knowledgeable. Public Health Inspector and Midwife are two good examples on whom to be acknowledged”*.[UMF1]

Overall, many participants criticised the attitudes of both the public and healthcare professionals, and it was suggested that Sri Lankans’ attitudes should be altered.

*“Indeed, the very first change which must take place is the change in people’s attitudes”*.[ULD1]

*“The problem lies not in raising awareness but in people’s attitudes. Even I am the same”*.[RVE1]

## 3. Discussion

This study investigated the KAPs of both human and animal healthcare professionals in Sri Lanka regarding antibiotic stewardship and AMR. To the best of our knowledge, this is the first qualitative study conducted on these topics in Sri Lanka, enabling a deeper exploration of the participants’ KAP than via quantitative surveys alone.

Healthcare professionals who participated in this study were familiar with the term “antibiotic resistance”. However, this study proposes that many healthcare professionals need to revise their antibiotic prescription behaviour to follow proper and updated guidance. To help them achieve this goal, it is recommended that deliberate and continuous professional development programmes should be established to update the understanding regarding how AMR develops and spreads among all groups and professions within the One Health umbrella. In addition, consideration of the education regarding AMR within pre-qualification teaching, including degrees and diploma studies, is warranted following the guidelines of the WHO [10].

The participants in this study regularly judged and blamed other professionals for their prescription behaviours, but often lacked self-awareness regarding their own poor behaviour. Some participants criticised other professionals for prescribing antibiotics to patients who had symptoms of viral infections. This behaviour may be due to a lack of basic antibiotic knowledge, or due to complexities in real-world practise. A common reason given for these inappropriate prescriptions was the diagnostic uncertainty of diseases and the prevention of secondary infections and complications [22,23].

The findings revealed that participants had poor comprehension of the development of AMR. Despite the fact that AMR evolves in bacteria, many of the participants in our study believed it began in the human or animal body. In addition, many participants did not understand the routes of AMR transmission within the ecosystem. Even though some participants were able to explain AMR transmission via animal-originated foods, soil and water, no one was able to explain the potential role of wild animals in AMR transmission in the ecosystem. In the context of ‘One Health’, poor understanding by participants of the development of AMR may lead them to underestimate the risks of AMR transmission between wild and farm animals, and between animals and humans [24,25].

As noted above, a lack of self-awareness regarding an individual’s gaps in knowledge and understanding was likely to be a major barrier to good antibiotic stewardship by healthcare professionals. Many participants commented on the behaviour of their clients that may contribute to the development and spread of AMR. These behaviours, by patients and livestock farmers, included the direct over-the-counter purchase of inappropriate antibiotics for self-medication and medication of farm and domestic animals, lack of adherence to the dosing schedules of prescribed antibiotics and the use of antibiotics prescribed for others. Similar problems of inappropriate usage of antibiotics by patients and farmers has been reported worldwide [26,27,28,29,30]. A recently published study described the factors affecting self-medication among the general public in Mozambique [31]. These factors included cultural beliefs about the healing power of antibiotics without the need for professional consultation, the easy availability of antibiotics, the easy access to a pharmacist’s advice, long wait times to access healthcare facilities, perceived poor quality of assistance at health care facilities and time and financial constraints. In the veterinary sector, many factors have been reported that lead to overuse and inappropriate use of antibiotics. For example, it requires less labour to give a whole herd antibiotic-medicated feed than provide individual treatment [32]. Concern for animal welfare and the wish to rapidly cure outbreaks of disease have led to antibiotic overuse among dairy farmers in New York State [33].

Both the human and veterinary healthcare participants in our study were under pressure from their clients to over-prescribe antibiotics. Antibiotic-demanding behaviour among clients has been reported in other studies [34,35,36]. Similar to our findings, antibiotics may be over-prescribed in order to secure and expand the client base of healthcare professionals [37,38]. For example, Australian veterinarians prescribed antibiotics to animal clients due to their fear of losing the business of their owners [39]. Another factor that leads to pressure for over-prescription of antibiotics by healthcare professionals in rural settings was that they are often dealing with a large area and a large population. This situation may be exacerbated by poor transport and diagnostic facilities in rural areas. In these circumstances, prescribers may think it ‘safer’ to provide antibiotics as a prophylactic measure rather than provide care tailored to the needs of individual patients. For example, an observational study reported that compared to urban women, rural women were more likely to receive inappropriately long courses of antibiotics for urinary tract infections [40]. Farmers in rural settings may over-treat their livestock without consulting a veterinarian in order to protect their animals and, ultimately, secure their livelihood [41]. Although economic-related factors affected the antibiotic prescribing behaviour of health professionals in low- and middle-income countries like Sri Lanka, this was not a problem among prescribers in one European country [42]. As identified in our study, a lack of diagnostic facilities acted as another barrier to good antimicrobial stewardship [43,44].

Similar to our findings, the influence of pharmaceutical companies was mentioned in other studies conducted in countries such as Nigeria, India and Uganda [45,46,47]. Participants reported that to achieve their market goals, pharmaceutical companies may pay commissions to healthcare professionals in Sri Lanka. A study conducted in Uganda described pharmaceutical company representatives offering prescribers incentives such as watches, clothes and even money to encourage them to prescribe their antibiotic as a drug of choice [47]. In comparison to over- or inaccurate prescribing due to poor understanding, this highlights some professionals prescribing for personal gain. Pharmaceutical companies have an important role to play in minimising the development and spread of AMR. Appropriate regulations and oversight are, however, needed to ensure that they do not contribute to the problem of AMR [48,49].

Antimicrobials used to treat infectious diseases in animals may be the same or similar to those used in people, and the ecosystem may link the development of AMR in humans and animals. Addressing the issue of AMR requires a One Health approach with multisector collaboration [50,51]; professions must work together. The results of our study suggest a lack of interconnection with other professionals and a tendency to blame other professionals for antibiotic misuse. Similarly, in another study, small companion animal veterinarians believed that antibiotic over-prescription in livestock contributed more to AMR than antibiotic use in companion animals [39]. Dairy veterinarians in New York State believed that their role in the growth of AMR was less than that of the medical sector [33]. Education and outreach programmes should be aimed at increasing interprofessional communication, increasing awareness of professions’ respective roles within a One Health approach to antimicrobial stewardship and reducing the tendency for some groups to blame others. The current study hopes to encourage future research regarding the specific development and evaluation of these interprofessional programmes.

This study has emphasised the importance of conducting AMR awareness programmes to educate the public and healthcare professionals. However, many medical participants in our study, despite showing several misunderstandings and prescribing misbehaviours with regard to AMR, thought that their knowledge was more than adequate and that they would not benefit from involvement in such programmes. Some general practitioners were unaware of new developments in understanding and prevention of AMR. It is clearly desirable that outreach programmes dealing with AMR are directed toward the medical profession. The results of our study also demonstrated the potential value of awareness programmes to enhance KAP on AMR among other healthcare workers. For example, field officers such as midwives, public health inspectors and livestock development officers who work closely with the community may play a significant role in knowledge dissemination among the wider community.

Where participants were aware of the ramifications of AMR on their own personal day-to-day lives, they spoke in a way which indicated sadness and a feeling of helplessness for their own patients. For professionals who spend their lives trying to help others, as indicated by the sub-category of ‘Understanding the patient and farmer’s perspectives’, this emotional impact of AMR may be significant. ‘Helplessness’ has been recently explored in this journal among patients who chose to self-medicate with antibiotics for protection against COVID-19 [52], and a study from India and South Africa reported the helplessness of lower professions in their interprofessional interactions during surgery regarding topics including antimicrobial stewardship [53]. However, it seems unusual for studies to consider these personal impacts for the professions at the front of the AMR battle, and this may be an area for future study to ensure the wellbeing of our professions.

This study has some limitations. It was conducted in two areas; therefore, the finding of our study cannot be generalised to the whole country or to other countries. However, the inclusion of both rural and urban participants enabled the results to include a range of views. Future quantitative studies may aim to establish the prevalence of the views identified within this study among healthcare professionals in Sri Lanka and other countries. Telephone interviews were conducted due to limitations on personal interactions imposed by COVID-19. Face-to-face interviews were the original preference of the project team, as they can facilitate the development of a good rapport between interviewer and interviewee as well as provide the ability to use visual cues to guide probing questions. However, telephone interviews are a well-used method in qualitative research, and studies have suggested that telephone interviews can produce the same results as face-to-face [54]. An independent researcher conducted the interviews, and while this meant that the project team were not able to ask questions during the interviews, it provided benefits including a lack of bias towards the responses and an ability to honestly ask basic questions due to a lack of prior understanding of AMR, which helped unpack the participants’ meanings. The use of a subject-specific novice who nevertheless has qualitative research experience in this way has been successfully used by some of the authors in previous research which identified that antibiotic stewardship was an area of concern for British equine practice.

## 4. Materials and Methods

### 4.1. Approach

A qualitative research approach was chosen because of its ability to offer in-depth explorations of complex phenomena that have not yet been discussed extensively in the literature. Furthermore, semi-structured interviews were conducted because of the possibility to explore the participants’ perceptions and let them speak freely. This study was approved by the institute of ethical clearance committee, Faculty of Medicine, University of Peradeniya, Sri Lanka. Ethics grant No. 2020/EC/28.

### 4.2. Participants

Non-probability sampling—namely, purposive sampling—was used to identify potential participants. The sampling aimed to achieve maximum variation based on the participant’s role and study site, as elaborated upon below. All potential participants lived and/or worked in the two study areas. All human and veterinary healthcare professionals from the locations were eligible to take part in the study. This included, for example, physicians, nurses, midwives and pharmacists (both degree and diploma pharmacists may legally practice in Sri Lanka) from the human healthcare service as well as veterinarians and livestock development instructors from the veterinary healthcare service. Two methods were used to identify interviewees. First, field-level government officers, mainly “Grama Niladhari” and “Smurdhi Development officers”, of the respective area were contacted by author YDG and through them, contact details (name, telephone number and address) of healthcare professionals were collected. Second, hospital websites and telephone directories were used to collect additional contact details. This study also embraced the snowball sampling technique to access possible interview partners as identified during the interview or recommended by the interviewees.

### 4.3. Field Context

Two areas of Sri Lanka were chosen for their distinct characteristics, particularly their urban or rural setting, to enhance diversity within the sample.

A one-square-kilometre region was identified as a study site from the urban area which was located in the Gampaha district of the Western province of Sri Lanka. Infrastructure for healthcare and the number of healthcare professionals and related workers in the urban area is comparatively higher than in the rural area. Another square kilometre area was identified from a rural area, which was located in the Badulla District of Uva province of Sri Lanka. Nearly all citizens living in the rural area belong to an indigenous tribal community called ‘Veddas’. The health system in the rural setting is enriched by a mix of Allopathic (conventional and modern medicine) and Ayurvedic (traditional medicine) systems of medicine that exist together; however, in the urban area, Allopathic medicine has become dominant and caters to the majority of the health needs of the people.

### 4.4. Data Collection Method

Author YDG contacted all participants and provided a formal introduction to the study and obtained informed consent. Contact details were then passed to an independent interviewer to arrange the interview. The interviewer was a trained researcher, female, holding a Bachelor of Arts in Sociology degree from the faculty of Arts, University of Peradeniya, Sri Lanka. She had experience in conducting health-related qualitative studies, but no prior experience with, or knowledge of antibiotics or AMR. To avoid potential bias of the interviewer during the interview, the interviewer was blinded to the profession of the participant.

An interview guide consisting of three main open-ended questions, and two to four prompts under each main question was developed by the research team. Overall, the questions aimed to provide data to answer the research question and were therefore focused on experiences with AMR and perceptions of ways to tackle AMR. Four telephone interviews were conducted as a pilot study and minor modifications were made as per the feedback of participants, including the addition of new prompts and edits to the order of the prompts to enable the desired exploration of the research question. The final interview guide can be seen in the Appendix A. Interviews were originally planned as face-to-face interviews in March 2021; however, they were affected by the COVID-19 pandemic and held as telephone interviews from July to August 2021. All interviews were conducted in Sinhalese, and the duration of the interviews ranged from 30 to 40 min. The interviews were audio-recorded and transcribed by the interviewer. Participants were offered the chance to check the content of their interviews; however, none chose to due to the time-consuming nature of the task. Author YDG checked the scientific accuracy of the transcription regarding AMR content by listening to the recordings. Participants were offered the chance to voluntarily withdraw from the study at any time; however, no participant requested to do so.

### 4.5. Data Analysis

Conventional qualitative content analysis techniques were used to analyse the data, as described by Hseih and Shannon (2005) [55]. In brief, all interviews were read as a whole to gain an initial impression of the concepts discussed. Data were then read in more detail to develop codes per word, line or paragraph, depending on the meaning. A code list was derived by authors YDG and RSK. Author ASF also reviewed transcripts and codes, and three researchers experienced in AMR collaborated in finalising the codes. After finalizing the codes, subcategories were developed to capture the collective meaning of a group of codes. Finally, depending on the relationships between subcategories, some were combined, leading to a final list of categories enabling a summary of the results to be reported as the findings of the study. Quotes from the transcripts were selected and utilised in the results section to highlight the dependability of the research. After the data analysis, the selected quotes were translated by an independent translator for the purpose of data interpretation and manuscript writing [56].

## 5. Conclusions

This study revealed some of the social dimensions of antibiotic usage and AMR in Sri Lanka. Healthcare professionals had a knowledge gap with regard to antibiotic stewardship and AMR. Interventions to enhance the quality of antibiotic prescribing and usage should be addressed both to health professionals and to the general public.

## Figures and Tables

**Table 1 antibiotics-11-00968-t001:** Demographic Characteristics of participants.

No	Site	Job Role (Bold Indicates Veterinary Healthcare Professionals)	Work Experience (Years)	Highest Educational Qualification	Participant Identity Code
1	Urban	Livestock Development Instructor	20	Diploma	ULD1
2	Urban	Veterinarian	15	Masters	UVE1
3	Urban	Veterinarian	16	Bachelor	UVE2
4	Urban	Veterinarian	13	Masters	UVE3
5	Urban	Livestock Development Instructor	23	Diploma	ULD2
6	Urban	Public Health Inspector	18	Diploma	UPH1
7	Urban	Veterinarian	6	Bachelor	UVE4
8	Urban	Nurse	10	Diploma	UNU1
9	Urban	Pharmacist	7	Diploma	UPM1
10	Urban	Dental Doctor	15	Bachelor	UDD1
11	Urban	Nurse	20	Diploma	UNU2
12	Urban	Physician	12	Bachelor	UHD1
13	Urban	Midwife	21	Diploma	UMF1
14	Urban	Physician	15	Bachelor	UHD2
15	Urban	Pharmacist	30	Diploma	UPM2
16	Urban	Nurse	25	Diploma	UNU3
17	Urban	Physician	2	Bachelor	UHD3
18	Urban	Nurse	4	Diploma	UNU4
19	Urban	Physician	30	Bachelor	UHD4
20	Urban	Medical Laboratory Technologist	32	Diploma	UML1
21	Rural	Veterinarian	12	Masters	RVE1
22	Rural	Physician	7	Bachelor	RHD1
23	Rural	Veterinarian	8	Bachelor	RVE2
24	Rural	Physician	4	Bachelor	RHD2
25	Rural	Livestock Development Instructor	2	Diploma	RLD1
26	Rural	Nurse	30	Diploma	RNU1
27	Rural	Midwife	10	Diploma	RMF1
28	Rural	Livestock Development Instructor	12	Diploma	RLD2
29	Rural	Pharmacist	25	Diploma	RPM1

**Table 2 antibiotics-11-00968-t002:** Categories and sub-categories of content analysis.

Category	Sub-Category
Understanding and observing AMR	AMR knowledge and experience acquisition
	Thoughts on AMR development and transition
Barriers to antimicrobial stewardship	Patient and farmer demands antibiotics
	Patient and farmer ability to self-treat and disregard healthcare advice
	Need to maintain professional status and client base
	Antibiotics as a commodity The issue of illegal prescription
Personal factors in, and as a result of, inappropriate antibiotic usage	Understanding the patient and farmer’s perspectives Consequences of the personal role in the AMR battle
How to tackle AMR	Thoughts on a personal role in tackling AMR
	Potential interventions to tackle AMR

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
