# Peer review of "Exploring Barriers to One Health Antimicrobial Stewardship in Sri Lanka: A Qualitative Study among Healthcare Professionals"

_antibiotics, 2022, doi:10.3390/antibiotics11070968_

Round 1
Reviewer 1 Report
This submitted manuscript titled “Exploring barriers to One Health antimicrobial stewardship in Sri Lanka: A qualitative study among healthcare professionals” investigated the Knowledge-Attitude-Practice (KAP) of both human (medical and dental) and animal healthcare professionals in Sri Lanka regarding antibiotic stewardship and Antimicrobial Resistance (AMR).
It is interesting as human and animal healthcare professionals are evaluated together, and this gives us chance to concentrate on AMR regardless of the specialisation of the professionals. Another point worth mentioning is the qualitative methodology used. It is not a methodology we are much accustomed to as health professionals. One Health approach of the World Health Organisation has been emphasised in addition to conducting AMR awareness programmes to educate people in general and healthcare professionals. The study has indicated the lack and therefore the need for an educational programme targeting not only all branches of health care professionals but also complementary allied health care professionals and related public or private sector workers, and in fact, including the target population (people, farmers etc), too.
There were surprising responses like one participant considering educating the public to be potentially dangerous [LINES 398-399]. While some participants could not explain the mechanism of AMR, or relate it to the human body instead of bacteria; some veterinarians emphasised the value of biosecurity management on livestock farms as a possible approach to combat AMR [LINE 386].
It is somewhat questionable if a moderator without prior knowledge about the subject (AMR, etc) could be really effective in conducting such an interview.
To summarise, the study reminds us that we have a lot to do to win this AMR battle.
I have indicated the need for minor corrections in spelling/English grammar:
-No need to abbreviate Sri Lanka as SL.
-The preferred verb tense suggested is “past tense”: As an example, “…The results of our study also demonstrate the potential…” [LINE 515]; the results of the study cannot be generalised as also mentioned by the authors in the LIMITATIONS section of the study, therefore past tense would be better. This is an example; please check this THROUGHOUT THE TEXT.
1. Uniformity in using UK English (UK English appears to have been preferred as it is the most used one in almost all instances):
1.a) “…Participants recognized poor behaviours as contributing to AMR…” [LINE 345]
1.b) “…antimicrobial stewardship awareness programs should be conducted to sensitize the general public and healthcare professionals…” [LINES 392-393]
2. English in general:
2.a) “…all those confines to ‘discussions’, just ‘discussions..." [LINE 373]
These are some examples; please check this THROUGHOUT THE TEXT.
-Data Availability Statement IS MISSING [LINES 639-643].
“…In this section, please provide details regarding where data supporting reported results can be found, including links to publicly archived datasets analyzed or generated during the study. Please refer to the suggested Data Availability Statements in the section “MDPI Research Data Policies” at https://www.mdpi.com/ethics. If the study did not report any data, you might add “Not applicable” here….” THIS IS THE TEXT OF THE TEMPLATE.
REFERENCES
Some references do not bear publication data. In such cases, at least, web links should be provided.
Ref#3 Organization, W.H. Antimicrobial stewardship programmes in health-care facilities in low-and middle-income countries: a WHO practical toolkit. 2019. [LINES 654-655]
Ref#4 Organization, W.H. Global action plan to control the spread and impact of antimicrobial resistance in Neisseria gonorrhoeae. 2012. LINES [656-657]
Ref#5 Lanka, S. National Strategic Plan for Combating Antimicrobial Resistance in Sri Lanka. 2017. [LINE 658]
Ref#10 Organization, W.H. Health workers’ education and training on antimicrobial resistance: curricula guide. 2019. [LINE 668]
Author Response
Thank you very much for your valuable comments. Please see the attachment

Reviewer 2 Report
attached

Author Response

(The authors gave the same response as above.)

Reviewer 3 Report
Great description and paper describing the survey and perceptions of some novel specialties in terms of AMR.
The manuscript is extremely long and tedious to read through. Globally, recommend that authors make the introduction more concise. Additionally, the results are very long - please consider making a table or figure to denote the questions asked and their results and refer to significant findings in the text.
For methods - please describe how the survey questions were formulated.
Also, please review for grammatical and spelling errors.
Author Response
Thank you very much for your valuable comments, please see the attachment

Reviewer 4 Report
Overall a well-written report, that, I must say, painted a rather shocking pictures. My recommendation is to accept the manuscript after minor corrections. I was really surprised of some of the findings that came out of this study. In particular, it was somewhat "disturbing" for me to read about lack of knowledge about AMR development (e.g. section 2.1.2) within healthcare professionals, and the pressure from pharmaceutical companies (or their representatives) to prescribe antibiotics at time where there may not have been a need. Reports like the one presented by the authors, really highlight that the road ahead for combating AMR is still very long.
Some suggestions/corrections are listed below:
1. lines 37-39: I suggest adding a comma: "regarding antibiotics and AMR, namely; physicians, veterinarians, dentists, nurses, pharmacists, medical laboratory technologists, public health inspectors, mid-wives and livestock development officers, were categorized under.."
2. line 52: change: "...blood, culture isolates..." to "blood and culture isolates..."
3. lines 92-95: avoid single sentence-paragraphs
4. line 257: correct capital letter after full stop and add comma: "...recommendations. however..." to : "...recommendations. However,...
4. lines 259-261, should be italic
